# Eye health and quality of life: an umbrella review protocol

Lama Assi ,[1] Lori Rosman,[2] Fatimah Chamseddine,[3] Perla Ibrahim,[4] Hadi Sabbagh,[4] Nathan Congdon,[5,6] Jennifer Evans,[7] Jacqueline Ramke ,[7,8] Hannah Kuper ,[9] Matthew J Burton,[7,10] Joshua R Ehrlich ,[11,12] Bonnielin K Swenor[1,13]

For numbered affiliations see end of article.

**Correspondence to**
Dr Bonnielin K Swenor;
bswenor@jhmi.edu

## ABSTRACT

**Introduction** Vision impairment and eye disease are major global health concerns and have been associated with increased morbidity and mortality, and lower quality of life. Quality of life, whether generic, vision-specific or disease-specific, is an important measure of the impact of eye health on people's daily activities, well-being and visual function, and is increasingly used to evaluate the impact of ophthalmic interventions and new devices. While many studies and reviews have examined the relationship between vision or eye health and quality of life across different contexts, there has yet to be a synthesis of the impact of vision impairment, eye disease and ophthalmic interventions on quality of life globally and across the lifespan.

**Methods and analysis** An umbrella review of systematic reviews will be conducted to address these two questions: (1) What is the association of vision impairment and eye disease with quality of life? (2) What is the impact of ophthalmic interventions on quality of life? A search of related literature will be performed on the 11 February 2020 in Medline Ovid, Embase.com, Cochrane Database of Systematic Reviews, Proquest Dissertations and Theses Global, and the grey literature, and repeated at the synthesis stage. Title/abstract and full-text screening, methodological quality assessment and data extraction will be conducted by reviewers working independently and in duplicate. Assessment of methodological quality and data extraction will be performed using Joanna Briggs Institute standard forms. Findings from the systematic reviews and their methodological quality will be summarised qualitatively in the text and using tables.

**Ethics and dissemination** No ethical approval is required. Results of this umbrella review will be published in a peer-reviewed journal and summarised in the *Lancet Global Health* Commission on Global Eye Health.

**Trial registration number** This protocol was registered in the Open Science Framework Registries (https://osf.io/qhv9g/).

## Strengths and limitations of this study

► The umbrella review approach allows for a comprehensive review of a very broad topic by summarising the evidence from multiple research syntheses into one systematic review of reviews.
► Study screening, critical appraisal and data collection will be conducted in duplicate.
► Standardised forms developed specifically for the conduct of umbrella reviews will be used for critical appraisal and data collection.
► Studies related to rare topics or special settings might not be included in systematic reviews, and thus would not be represented in this umbrella review, a main limitation of our work.
► Only systematic reviews published in English will be included.

that year.[3] Despite reductions in age-specific prevalence, the number of people with vision impairment and blindness is projected to increase due to population growth and ageing.[2] Vision impairment is associated with negative health outcomes, such as having multiple chronic conditions,[4 5] and increased mortality,[6] and also induces substantial socioeconomic consequences for individuals,[7] and an associated lower quality of life.

Objective clinical measures, like visual acuity, intraocular pressure or fundus imaging, are widely used in the clinical and research settings to assess eye health, but often fail to capture the impact of vision impairment or eye disease on individuals' daily activities or social well-being.[8] Quality-of-life instruments, on the other hand, measure patient-reported outcomes, such as perceived health, physical, mental, emotional or social well-being, and even vision-specific function. These measures are important as vision impairment can have a large impact on quality of life, possibly to an even greater degree than major conditions such as stroke, heart disease or diabetes.[9] Both severe conditions that lead to marked reduction in vision like age-related macular degeneration,[10] and

## INTRODUCTION

Vision impairment is a major cause of disability worldwide.[1] In 2015, an estimated 36 million people were blind, 217 million had moderate or severe vision impairment and over a billion people experienced near-vision impairment (presbyopia).[2] Cataract and uncorrected refractive error are correctable conditions which accounted for 78% of global visual impairment

highly symptomatic conditions which may not be associated with impaired vision, like dry eye syndrome,[11] have been associated with decreased health-related quality of life.

The use of quality-of-life instruments has gained popularity in ophthalmic studies, including clinical trials, over the past decade.[12] While there is a wide range of quality-of-life instruments available, vision-related quality-of-life instruments are frequently used in ophthalmic studies, as these questionnaires are more sensitive to the impact of subtle vision changes on daily function compared with more general health-related or generic quality-of-life tools.[13] Reduced visual acuity[14] and visual field loss[15] are both associated with worsening in vision-related quality of life; glaucoma[16–19] and cataract[16 19] are associated with worse vision-related function, independent of visual acuity. In ophthalmic clinical trials, health-related, vision-related and even disease-specific scales have been used as secondary outcome measures, and more recently, as primary outcomes as well.[20 21] Patient-reported outcomes are also increasingly being incorporated in the evaluation of new ophthalmic devices, and the Food and Drug Administration even provides guidance on using them to support labelling claims too.[22 23]

There has yet to be a global assessment of the impact of eye health, including vision impairment, eye disease and ophthalmic interventions on quality of life across the lifespan, despite the growing number of ophthalmic studies assessing quality of life, and increased value placed on patient-reported outcomes. Prior studies on vision and quality of life have usually focused on specific countries (eg, USA,[24] Finland,[25] South Korea,[26] Nigeria[27]), populations (eg, Malay population in Singapore,[28] Latino population[29] and indigenous peoples of the Americas[30] in the USA) or settings (eg, community,[31] outpatient clinics[32]). Even reviews summarising the evidence about the impact of vision on quality of life have usually focused on specific age groups (eg, children,[33 34] older adults[35]), eye conditions (eg, glaucoma,[36] diabetic retinopathy,[37] dry eye[38]) or interventions (eg, low vision rehabilitation for children,[34] anti-vascular endothelial growth factor therapy for age-related macular degeneration[39]).

This umbrella review (or systematic review of systematic reviews) will examine the impact of vision impairment, eye disease and ophthalmic interventions on quality of life globally and across the lifespan. An umbrella review approach allows us to maintain a broad scope while relying on the highest quality of evidence, given the large number of primary studies[9–11 14–19 24–32] and reviews on this topic.[34–41] A search of three systematic review registries (the International Prospective Register of Systematic Review [PROSPERO], Joanna Briggs Institute Systematic Review Register, and Open Science Framework Registries) has shown that there is currently no systematic or umbrella review underway for this topic.

## Objectives and questions

This umbrella review of systematic reviews aims to identify and synthesise currently available knowledge about the association of vision and eye disease with quality of life on a global level. Two questions will be addressed:

1. What is the association between vision impairment or eye disease and quality of life?
2. What is the impact of ophthalmic interventions on quality of life?

## METHODS AND ANALYSIS

This protocol was registered in the Open Science Framework Registries (https://osf.io/qhv9g/). It was designed by following the Joanna Briggs Institute guidelines for the conduct and preparation of umbrella reviews,[42] and the Preferred Reporting Items for Systematic review and Meta-Analysis Protocols guidelines for the reporting of systematic review protocols (online supplementary file 1).[43] The anticipated start date of this study is 11 February 2020. Any changes to the methodological approach will be dated and described in detail in the final umbrella review report.

### Inclusion criteria

Systematic reviews and meta-analyses that evaluate the impact of vision impairment, eye disease or ophthalmic interventions on quality of life will be included in this umbrella review.

### Types of participants

Systematic reviews of studies with participants who have vision impairment or an eye disease will be included. Vision impairment can be self-reported or assessed objectively, using any measure of visual function, including, but not limited to, visual acuity (corrected or uncorrected, distance or near), contrast sensitivity or visual fields. Eye disease diagnosis can be based on self-report, medical chart or claims data or an objective assessment of symptoms, clinical signs or imaging findings. Eye diseases that will be explored include, but are not limited to, the WHO priority eye diseases, which are the most common causes of vision impairment worldwide: cataract, onchocerciasis, trachoma, refractive errors, age-related macular degeneration, diabetic retinopathy, glaucoma, corneal opacities, childhood blindness and genetic eye diseases.[44]

Systematic reviews with sample populations from any age group (children, working-age adults or older adults), country (low, middle or high income) or setting (community, hospital, clinic, institution) will be included.

### Interventions

Systematic reviews that examine interventions will also be included and will help answer the second question specifically, the impact of ophthalmic interventions on quality of life. The comparison group can be the same group preintervention, another group that receives another intervention, or another group receiving no intervention. Any ophthalmic intervention identified will be included, as long as its main aim is to correct or improve vision, slow down the progression of vision loss, improve functional ability among those with vision loss (eg, low vision

rehabilitation, use of assistive devices) or relieve eye pain or discomfort.

## Outcomes

Studies that measure any aspect of quality of life (generic or health-related, vision-related or disease-specific) will be included. Studies can report on one or all domains used to measure quality of life. Systematic reviews of both quantitative and qualitative studies are eligible for inclusion. Examples of quality-of-life instruments are the WHO Quality of Life Assessment Instrument (health-related quality of life), the National Eye Institute Vision Function Questionnaire (vision-related quality of life) and the Catquest-9SF questionnaire (cataract-specific quality of life).

## Types of studies

Only systematic reviews (with or without and meta-analyses) are eligible for inclusion. A systematic review will be defined as a review that includes every one of these items: a research question, a search strategy with the sources searched, inclusion and exclusion criteria, screening methods, a discussion about the quality of included studies and risk of bias and information about data analysis and synthesis.[45] Systematic reviews of observational and interventional studies will be included, but those that incorporate case series or expert opinion as their source of evidence will be excluded. All other types of reviews including narrative reviews and scoping reviews will be excluded.

## Search strategy

An academic librarian developed a comprehensive search strategy based on similar ones used by Cochrane Eyes and Vision Group. Search strategies were developed using a combination of controlled vocabulary and keywords to represent vision terms, eye diseases, including all the WHO priority eye disease listed above, and ophthalmic interventions, as well as search terms for quality of life, including some commonly used scales, and terms to identify systematic reviews (see online supplementary file 2 for a detailed search strategy).

The following databases will be searched on 11 February 2020: Medline Ovid (1946 to present), Embase.com (1947 to present), Cochrane Database of Systematic Reviews (1995 to present), Proquest Dissertations and Theses Global (1861 to present). A search for grey literature will include sources such as reports from governments and non-governmental organisations, and databases including the Open Grey and the Agency for Healthcare Research and Quality. The search will be limited to articles published in English with no restrictions on the year of publication. We will search the references of included studies for additional systematic reviews.

The search will be run again in the synthesis stage to identify any relevant reviews published since the initial search.

## Study selection

Citations retrieved from the searches will be imported to Endnote, where any duplicates will be removed. Then, references will be imported to Covidence, a web-based software platform that streamlines the production of systematic reviews. Four reviewers will work in pairs to screen the studies independently and in duplicate. Conflicts will be discussed in the presence of a third reviewer from the second pair; if no consensus is reached, the senior author will be consulted.

Review selection will take place at the level of title/abstract and full text. Reviews judged to be of insufficient quality for inclusion will also be excluded at the methodological assessment stage. Articles will first be screened at the level of the titles and abstracts. At this stage, all systematic reviews and meta-analyses, published in English, and that address a vision-related topic (vision impairment, eye disease or ophthalmic intervention) and quality of life will be included. Articles that are identified as systematic reviews in the title or abstract, using the terms 'systematic review' or 'meta-analysis', will be included. Reviews that are not explicitly identified as such will be moved to full-text review if the methods suggest they may be systematic reviews based on the definition used above. Reviews that are clearly not related to quality of life will be excluded at this stage, but reviews of interventional studies that do not specifically address quality of life in their title or abstract will go to full-text review, as quality of life may be a secondary outcome that is only mentioned in the text.

In the next step, the included reviews will undergo full-text screening and will be included if they meet the criteria listed above for a systematic review, if none of their primary studies are case series/case reports or expert opinion, and if they specifically assess the impact of the vision-related topic on quality of life. The final study selection will be made after assessment of methodological quality. Reasons for exclusion will be logged.

## Assessment of methodological quality

Systematic reviews that are deemed eligible for inclusion will be assessed for their methodological quality using the Joanna Briggs Institute Critical Appraisal Checklist for Systematic Reviews and Research Syntheses.[46] The four reviewers will work again in pairs to do the assessment independently and in duplicate. Joanna Briggs Institute SUMARI, a web-based review software that has partnered with Covidence, will be used to facilitate the critical appraisal step.

The Joanna Briggs Institute Critical Appraisal Checklist contains 11 items related to systematic review methodology, each graded as 'Yes', 'No', 'Unclear' or 'Not Applicable'. It can be used to appraise both quantitative and qualitative systematic reviews. The form will be piloted by the four reviewers by testing it on two studies before starting independent appraisals; these reviewers will compare their results and discuss what constitutes an acceptable level of information to decide if a review meets or does not meet the criteria, and when is it 'Unclear'.

Systematic reviews for which at least one of the items 'clear review question', 'appropriate inclusion criteria', 'appropriate search strategy', or 'appropriate criteria for critical appraisal' are graded as 'Unclear' or 'No' will be considered to be of insufficient quality for inclusion, and as such would be excluded at this stage.

Results of the quality appraisal for each review will be presented in a table, and the overall methodological quality of the included reviews will be summarised in the text.

### Data collection

Data will be extracted from the final list of articles included using the Joanna Briggs Institute Data Extraction Form for Review for Systematic Reviews and Research Syntheses (see online supplementary file 3 for a blank copy of the sample data extraction sheet).[42] Again, the four reviewers will work in pairs to extract the data independently and in duplicate, using the Joanna Briggs Institute SUMARI software.

In brief, the standardised form will be used to extract information about citation details (eg, author, year), systematic review methodology (eg, objectives, participants, setting/context, search strategy, appraisal instrument used), characteristics of the included studies (eg, date range, number and types of studies, country of origin, rating of their quality, outcomes reported) and findings of the systematic review (eg, method of synthesis/analysis employed, results/findings). Additionally, information about the review's funder or sponsor and their role, when applicable, and the reviewers' overall assessment of the quality of the evidence, such as Grading of Recommendations Assessment, Development and Evaluation (GRADE), will be collected. The GRADE quality assessment is based on the primary studies' quality and design, and the consistency and directness of the findings.[47]

In the comments section, the following information will be indicated: (1) if the review is about vision impairment/ eye health or an ophthalmic intervention, (2) the functional vision measure used or eye disease or intervention, (3) if the population belongs to a specific age group, country income group or setting. This will help classify the reviews for the synthesis.

Before the reviewers start collecting data independently, the form will be piloted. All four reviewers will extract data from two articles, compare their answers and discuss them to ensure that they all interpret the questions in the same way.

### Data summary

A qualitative synthesis of the findings will be presented in the text and using tables describing study characteristics and the overall umbrella review results (summary of findings). When presenting study characteristics, studies will first be divided according to the question they address: the impact of vision impairment or eye disease on quality of life, or the impact of ophthalmic interventions on quality of life. Each question will have a table for the quantitative systematic reviews, and another one for the qualitative systematic reviews, as the study characteristic information reported/presented for each is different. Within each table, results will be stratified according to the outcome measured (health-related, vision-related or condition-specific quality of life), and functional vision measured, eye disease or intervention (figure 1).

Study characteristics tables for the quantitative systematic reviews will include the following information: the number of studies in the systematic review, the number of participants from the included studies, estimates computed and the heterogeneity of the results. For qualitative systematic reviews, the final synthesised findings will be presented along with information about the study context. Overlaps of original research studies in the included systematic reviews will be presented.

Summary of findings tables will present an overall summary for each question, exposure, outcome and,

| Level 1:<br>Question | Level 2:<br>Study Type | Level 3:<br>Outcome | Level 4:<br>Exposure | Level 5:<br>Population |
|---|---|---|---|---|
| • Question 1: Vision impairment or eye disease<br>• Question 2: Ophthalmic interventions | • Quantitative systematic reviews<br>• Qualitative systematic reviews | • Health-related quality of life<br>• Vision-related quality of life<br>• Condition-specific quality of life | • Functional vision measure (Question 1)<br>• Eye disease (Question 1)<br>• Intervention (Question 2) | • Potential subgroups for summary of findings tables:<br>• Age group<br>• Country income level<br>• Setting |

**Figure 1** Organisation of findings. Reviews will be divided according to the question they address (question 1 being vision impairment/eye disease, and question 2, ophthalmic interventions). For each question, reviews will be categorised as quantitative or qualitative, and within each category, they will be further grouped based on their quality-of-life measure and exposure (specific functional vision measure or eye disease for question 1, and intervention for question 2). Summary of findings tables will be further stratified by study population depending on the results available; potential subgroups include age category (children, working-age adults or older adults), country (low, middle or high income) or setting (community, hospital, clinic, institution).

 Assi L, et al. BMJ Open 2020;**10**:e037648. doi:10.1136/bmjopen-2020-037648

when applicable, subgroup (age, country income group, setting; figure 1), along with an assessment of the strength of the evidence for each finding, such as GRADE, when included in the review.

## Study limitations

The study methodology has some limitations that may impact the final results of the review. Using the umbrella review approach limits the results to those found in articles that have been included in systematic reviews. While this means that studies about rare diseases or topics that have not yet been addressed by systematic reviews will not be included, it is this approach that will allow us to perform a global assessment of a broad topic in a systematic manner. Moreover, using strict criteria to define a systematic review, and limiting inclusion to those that meet certain quality requirements may further decrease the number of studies included, but it will allow us to focus on the available high-quality evidence. In regard to the first question, vision impairment and eye diseases may be defined and diagnosed differently in each review, thus making it harder to combine the evidence. Likewise, a large number of interventions may be identified for the second question, and the type of comparison groups might differ between reviews (intervention compared with no intervention or intervention compared with another intervention), making the synthesis of the evidence challenging. However, using the methods detailed above to present the results and summarise the findings will allow us to organise the findings in a systematic manner and present enough context for the reader to interpret the results. Finally, as with any umbrella review, there may be overlap in the primary studies included in each systematic review; however, we will highlight any overlap of studies in the tables.

## Patient and public involvement

There is no patient or public involvement in this study.

## ETHICS AND DISSEMINATION

Only published studies will be examined for this systematic review; therefore, no ethical approval is required. If any changes to the protocol are made, they will be described in the final umbrella review report.

Results from this study will be published in a peer-reviewed journal and summarised in *The Lancet Global Health* Commission on Global Eye Health.

## Author affiliations

[1]Wilmer Eye Institute, Johns Hopkins University School of Medicine, Baltimore, Maryland, USA
[2]Welch Medical Library, Johns Hopkins University School of Medicine, Baltimore, Maryland, USA
[3]Clinical Research Institute, American University of Beirut Faculty of Medicine, Beirut, Lebanon
[4]Department of Ophthalmology, American University of Beirut Faculty of Medicine, Beirut, Lebanon
[5]Centre for Public Health, Queen's University Belfast School of Medicine Dentistry and Biomedical Sciences, Belfast, UK
[6]Zhongshan Ophthalmic Center, Sun Yat-Sen University, Guangzhou, China
[7]International Centre for Eye Health, London School of Hygiene & Tropical Medicine, London, UK
[8]School of Optometry and Vision Science, The University of Auckland, Auckland, New Zealand
[9]International Centre for Evidence in Disability, London School of Hygiene & Tropical Medicine, London, UK
[10]Cornea & External Eye Disease, Moorfields Eye Hospital, London, UK
[11]Department of Ophthalmology and Visual Sciences, University of Michigan, Ann Arbor, Michigan, USA
[12]Institute for Healthcare Policy and Innovation, University of Michigan, Ann Arbor, Michigan, USA
[13]Department of Epidemiology, Johns Hopkins University Bloomberg School of Public Health, Baltimore, Maryland, USA

**Contributors** BKS is the guarantor of the review. All authors contributed to the development of the selection criteria, the risk of bias assessment strategy, data extraction criteria and approved the final manuscript. LR developed the search strategy. NC, JRE, JR, HK, MJB and JE provided expertise on systematic and umbrella review methodology, and global eye health. LA, LR and BKS drafted the manuscript. FC, PI, HS, NC, JRE, JR, HK, MJB and JE revised the manuscript critically for important intellectual content.

**Funding** BKS is supported by the National Institutes of Health (NIA K01AG052640). MJB is supported by the Wellcome Trust (207472/Z/17/Z). JR is a Commonwealth Rutherford Fellow, funded by the UK government through the Commonwealth Scholarship Commission in the UK. JRE is supported by the National Institutes of Health (K23EY027848). The Lancet Global Health Commission on Global Eye Health is supported by The Queen Elizabeth Diamond Jubilee Trust, Moorfields Eye Charity (grant number GR001061), NIHR Moorfields Biomedical Research Centre, The Wellcome Trust, Sightsavers, The Fred Hollows Foundation, The SEVA Foundation, The British Council for the Prevention of Blindness and Christian Blind Mission. No funder had any role in the design or conduct of this work.

**Competing interests** None declared.

**Patient and public involvement** Patients and/or the public were not involved in the design, or conduct, or reporting, or dissemination plans of this research.

**Patient consent for publication** Not required.

**Provenance and peer review** Not commissioned; externally peer reviewed.

**ORCID iDs**
Lama Assi http://orcid.org/0000-0001-5855-3896
Jacqueline Ramke http://orcid.org/0000-0002-5764-1306
Hannah Kuper http://orcid.org/0000-0002-8952-0023
Joshua R Ehrlich http://orcid.org/0000-0002-0607-3564

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
