## [Reviewer comments · BMJ Open]

ARTICLE DETAILS

TITLE (PROVISIONAL)	Eye Health and Quality of Life: An Umbrella Review Protocol
AUTHORS	Assi, Lama; Rosman, Lori; Chamseddine, Fatimah; Ibrahim, Perla; Sabbagh, Hadi; Congdon, Nathan; Evans, Jennifer; Ramke, Jacqueline; Kuper, Hannah; Burton, Matthew J; Ehrlich, Joshua; Swenor, B

VERSION 1 – REVIEW

REVIEWER	Mehul C Mehta, MD Department of Ophthalmology Harvard Medical School, Boston USA
REVIEW RETURNED	08-May-2020

GENERAL COMMENTS	This is an important and well-defined protocol for an Eye Health & Quality Of Life (QOL) umbrella review, aimed at addressing a set of three critical questions, all of which are important to comprehensively understand the impact of eye diseases on QOL. The question around the impact of ophthalmic interventions on QOL can also serve as the basis to further understand the 'value' of the eye care delivered, when combined with the economics of care delivery. The temporal QOL question that this protocol envisions to explore is particularly interesting especially if it is possible to compare interventions versus none, over time, though this will probably be the most challenging component of the study to comprehensively and accurately assess. I do however suggest that the authors consider outlining the limitations of their umbrella review protocol in the context of each of their three study questions. I look forward to the completion of this study and the results.
---

REVIEWER	Rosie Gilbert 1. Addenbrooke's Hospital, Cambridge University Hospitals NHS FT, UK 2. Moorfields Eye Hospital, City Road, Moorfields Eye Hospital NHS FT, UK
REVIEW RETURNED	11-May-2020

GENERAL COMMENTS	This is a useful and important topic for review. It is good to see that the authors are investigating the association between eye disease and quality of life (QoL), as well as vision impairment and QoL. Chronic/ recurrent anterior uveitis is another example of an eye disease which does not necessarily impact vision but has significant impact on QoL.
---

REVIEWER	Dr Elesh Kumar Jain Sadguru Netra Chikitsalaya, Shri Sadguru Seva Sangh Trust
-----------------	--

	Madhya Pradesh , India
REVIEW RETURNED	15-May-2020

GENERAL COMMENTS	Overall it is a well-planned study. One noted point is that; authors have mentioned about the piloting of the form (Data collection tool) by testing it on two studies before starting independent. The author could have actually done the pilot at first instance and then finalize the study protocol. This way the team would have also ensured the method they have chosen (four reviewers will work in pairs to extract the data independently and in duplicate) is also reliable or modifications in the same is required to make it more efficient
--

REVIEWER	Giuseppe Biondi-Zoccai Sapienza University of Rome, Latina, Italy
REVIEW RETURNED	01-Jun-2020

GENERAL COMMENTS	I enjoyed reading this protocol for an ophthalmologic umbrella review, which duly follows the practices I best support (eg Biondi-Zoccai G, editor. Umbrella Reviews: Evidence Synthesis With Overviews of Reviews and Meta-epidemiologic Studies, Springer, 2016). I only have the following minor suggestions: 1. I tried to retrieve the protocol from OSF but found it difficult. Probably you could improve indexing there. 2. Careful readers as well as newbies could benefit from a blank sample data extraction sheet, which they could use in their own umbrella reviews. 3. While I understand that quantitative synthesis is beyond the scope of your main analysis, network meta-analysis and multivariate meta-analysis could be useful to summarize more poignantly very strong findings. 4. While uncommon, some form of small study effect/publication bias analysis could be attempted. What's your take on this?
--

VERSION 1 – AUTHOR RESPONSE

Reviewer(s)' Comments to Author:

Reviewer: 1

Reviewer Name

Mehul C Mehta, MD

Institution and Country

Department of Ophthalmology Harvard Medical School, Boston USA

Please state any competing interests or state 'None declared':

None Declared

Please leave your comments for the authors below

This is an important and well-defined protocol for an Eye Health & Quality Of Life (QOL) umbrella review, aimed at addressing a set of three critical questions, all of which are important to comprehensively understand the impact of eye diseases on QOL. The question around the impact of ophthalmic interventions on QOL can also serve as the basis to further understand the 'value' of the eye care delivered, when combined with the economics of care delivery. The temporal QOL question that this protocol envisions to explore is particularly interesting especially if it is possible to compare

interventions versus none, over time, though this will probably be the most challenging component of the study to comprehensively and accurately assess. I do however suggest that the authors consider outlining the limitations of their umbrella review protocol in the context of each of their three study questions. I look forward to the completion of this study and the results.

Thank you for your comments! We have added a new section “Study Limitations” at the end of the protocol. It is copied here:

The study methodology has some limitations that may impact the final results of the review. Using the umbrella review approach limits the results to those found in articles that have been included in systematic reviews. While this means that studies about rare diseases or topics that have not yet been addressed by systematic reviews will not be included, it is this approach that will allow us to perform a global assessment of a broad topic in a systematic manner. Moreover, using strict criteria to define a systematic review, and limiting inclusion to those that meet certain quality requirements may further decrease the number of studies included, but it will allow us to focus on the available high-quality evidence. In regard to the first question, vision impairment and eye diseases may be defined and diagnosed differently in each review, thus making it harder to combine the evidence. Likewise, a large number of interventions may be identified for the second question, and the type of comparison groups might differ between reviews (intervention compared to no intervention, or intervention compared to another intervention), making the synthesis of the evidence challenging. However, using the methods detailed above to present the results and summarize the findings will allow us to organize the findings in a systematic manner and present enough context for the reader to interpret the results. Finally, as with any umbrella review, there may be overlap in the primary studies included in each systematic review; however, we will highlight any overlap of studies in the tables.

Reviewer: 2

Reviewer Name

Rosie Gilbert

Institution and Country

1. Addenbrooke's Hospital, Cambridge University Hospitals NHS FT, UK

2. Moorfields Eye Hospital, City Road, Moorfields Eye Hospital NHS FT, UK

Please state any competing interests or state 'None declared':

None declared

Please leave your comments for the authors below

This is a useful and important topic for review. It is good to see that the authors are investigating the association between eye disease and quality of life (QoL), as well as vision impairment and QoL. Chronic/ recurrent anterior uveitis is another example of an eye disease which does not necessarily impact vision but has significant impact on QoL.

Thank you for your comment! We moved forward with the review and uveitis is certainly included!

Reviewer: 3

Reviewer Name

Dr Elesh Kumar Jain

Institution and Country

Sadguru Netra Chikitsalaya, Shri Sadguru Seva Sangh Trust

Madhya Pradesh , India

Please state any competing interests or state 'None declared':
None Declared

Please leave your comments for the authors below

Overall it is a well-planned study. One noted point is that; authors have mentioned about the piloting of the form (Data collection tool) by testing it on two studies before starting independent. The author could have actually done the pilot at first instance and then finalize the study protocol. This way the team would have also ensured the method they have chosen (four reviewers will work in pairs to extract the data independently and in duplicate) is also reliable or modifications in the same is required to make it more efficient

Thank you for your comment! We agree that this would have helped us ensure that we have the right approach from the start. We already moved forward with the review and started the data extraction process, but this is definitely a point to keep in mind for future works.

Reviewer: 4

Reviewer Name

Giuseppe Biondi-Zoccai

Institution and Country

Sapienza University of Rome, Latina, Italy

Please state any competing interests or state 'None declared':

None declared

Please leave your comments for the authors below

I enjoyed reading this protocol for an ophthalmologic umbrella review, which duly follows the practices I best support (eg Biondi-Zoccai G, editor. Umbrella Reviews: Evidence Synthesis With Overviews of Reviews and Meta-epidemiologic Studies, Springer, 2016).

I only have the following minor suggestions:

1. I tried to retrieve the protocol from OSF but found it difficult. Probably you could improve indexing there.

Thank you for pointing this out! We just realized that the protocol was not made public. This was fixed and the doi was changed in the abstract and text.

This is the new doi: <https://osf.io/qhv9g/>

2. Careful readers as well as newbies could benefit from a blank sample data extraction sheet, which they could use in their own umbrella reviews.

Thank you for your comment. We agree and we have added it to the Online Supplemental File 3.

3. While I understand that quantitative synthesis is beyond the scope of your main analysis, network meta-analysis and multivariate meta-analysis could be useful to summarize more poignantly very strong findings.

Thank you for your comment. We agree that a network meta-analysis can summarize results more strongly, however, as you mentioned, this is beyond the scope of our analysis. Since we are doing a global review of a very broad topic, we expect to find a large number of different exposures and interventions to evaluate. However, we will keep in mind these approaches for future, more focused, work.

4. While uncommon, some form of small study effect/publication bias analysis could be attempted. What's your take on this?

While we do recognize that our results will be affected by publication bias, we believe that with such a broad topic, and without a network meta-analysis, publication bias analysis is beyond the scope of our analysis. We hope to at least minimize publication bias by searching several electronic databases as well as some grey literature (Open Gray, Agency for Healthcare Research and Quality). Moreover, we will be able to know which systematic reviews assessed the likelihood of publication bias at the critical appraisal step using the JBI Critical Appraisal Checklist.

VERSION 2 – REVIEW

REVIEWER	Giuseppe Biondi-Zoccai Sapienza University of Rome, Latina, Italy
REVIEW RETURNED	30-Jun-2020
GENERAL COMMENTS	All my comments have been satisfactorily addressed